

# Screening and identification of NOTCH1, CDKN2A, and NOS3 as differentially expressed autophagy-related genes in erectile dysfunction

Chao Luo[1,*], Xiongcai Zhou[1,2,*], Li Wang[3], Qinyu Zeng[1], Junhong Fan[1], Shuhua He[1], Haibo Zhang[1] and Anyang Wei[1]

[1] Department of Urology, Nanfang Hospital, Southern Medical University, Guangzhou, Guangdong, China
[2] Department of Urology, Guangzhou Eighth People's Hospital, Guangzhou Medical University, Guangzhou, Guangdong, China
[3] School of Basic Medical Science, Southern Medical University, Guangzhou, Guangdong, China
* These authors contributed equally to this work.

Corresponding authors
Haibo Zhang, hai516@163.com
Anyang Wei, profwei@126.com

## ABSTRACT

**Background**. Loss of function of key autophagy genes are associated with a variety of diseases. However specific role of autophagy-related genes in erectile dysfunction ED remains unclear. This study explores the autophagy-related differentially expressed genes (ARGs) profiles and related molecular mechanisms in Corpus Cavernosum endothelial dysfunction, which is a leading cause of ED.

**Methods**. The Gene Expression Omnibus (GEO) database was used to identify the key genes and pathways. Differentially expressed genes (DEGs) were mined using the limma package in R language. Next, ARGs were obtained by matching DEGs and autophagy-related genes from GeneCard using Venn diagrams. Gene Ontology (GO) and Kyoto Encyclopedia of Genes and Genomes (KEGG) analyses of ARGs were described using clusterProfiler and org.Hs.eg.db in R. Moreover, hub ARGs were screened out through protein-protein interaction (PPI), gene-microRNAs, and gene-transcription factors (TFs) networks then visualized using Cytoscape. Of note, the rat model of diabetic ED was established to validate some hub ARGs with qRT-PCR and Western blots.

**Results**. Twenty ARGs were identified from four ED samples and eight non-ED samples. GO analysis revealed that molecular functions (MF) of upregulated ARGs were mainly enriched in nuclear receptor activity. Also, MF of downregulated ARGs were mainly enriched in oxidoreductase activity, acting on NAD(P)H and heme proteins as acceptors. Moreover, six hub ARGs were identified by setting high degrees in the network. Additionally, hsa-mir-24-3p and hsa-mir-335-5p might play a central role in several ARGs regulation, and the transcription factors-hub genes network was centered with 13 ARGs. The experimental results further showed that the expression of Notch1, NOS3, and CDKN2A in the diabetic ED group was downregulated compared to the control.

**Conclusions**. Our study deepens the autophagy-related mechanistic understanding of endothelial dysfunction of ED. NOTCH1, CDKN2A, and NOS3 are involved in the regulation of endothelial dysfunction and may be potential therapeutic targets for ED by modulating autophagy.

# INTRODUCTION

Erectile dysfunction (ED), also known as impotence, is the inability to maintain enough erections to achieve the satisfactory sexual activity (*Muneer et al., 2014*). ED is a common disease with high incidence worldwide, and there are few therapeutic methods. It is associated with aging, diabetes, and postoperative complications. The etiology of ED is mainly vascular, caused by nitric oxide (NO) metabolism disorder in the cells. NO enhancing drugs like phosphodiesterase 5 inhibitors (PDE-5i), which alleviate NO's decay, are widely utilized but still fail sometimes (*Altabas & Altabas, 2015*). The dysfunction of endothelial cells (ECs) plays a vital role in the pathophysiological process of ED. The complex pathogenesis of diabetic ED, including nerves, blood vessels, endothelium, smooth muscle, other tissues, and organs, have been long studied. As an independent factor, hyperglycemia leads to vascular endothelial cell injury and dysfunction. Vascular endothelial integrity and barrier function are destroyed through oxidative stress injury, leading to vascular disease.

Autophagy is an evolutionarily conserved eukaryotic self-catabolic mechanism. It balances energy sources, clears misaggregated or folded proteins, and clears damaged organelles (such as mitochondria, endoplasmic reticulum, and peroxisome). It is involved in many biological functions including development, in response to nutrient stress, cellular differentiation, and resistance to pathogens. Therefore, autophagy serves as a survival mechanism, although its deregulation is a related to non-apoptotic cell death. There is growing evidence that mutations or suppression of key autophagy genes are associated with cancer, neuropathy, cardiovascular disease, autoimmune diseases, and other diseases (*Glick, Barth & Macleod, 2010*; *Levine & Kroemer, 2019*). Studies have confirmed that endothelial autophagy plays a critical role in the occurrence and development of cavernous endothelial cell lesions in ED (*Lin et al., 2018*; *Zhang et al., 2019a*). However, the role and specific regulatory mechanism of the autophagy gene in ECs remains unclear.

In this study we expected to analyze the human ED dataset in the GEO database using bioinformatics to explore the autophagy-related genes and related molecular mechanisms connected with endothelial dysfunction. Then, NOTCH1, PPARG, NOS3, KEAP1, CDKN2A, and IRS1 were predicted to affect endothelial dysfunction of ED as autophagy-related hub genes. Additionally, NOTCH1, CDKN2A, and NO3 were validated by establishing rat model of diabetic ED (DMED), and may be the underlying target molecules to treat ED by modulating autophagy.

# MATERIALS & METHODS

## ED datasets

Microarray data were obtained from the Gene Expression Omnibus (GEO) database (http://www.ncbi.nlm.nih.gov/geo) (*Edgar, Domrachev & Lash, 2002*). Specifically, we downloaded the gene expression profile of GSE10804 (Affymetrix GPL571platform,
**Table 1  Erectile dysfunction datasets ( GSE10804).**

| Simple | Origin | Source name | Cell type | Used in comparison |
|---|---|---|---|---|
| GSM272854 | corpus cavernosum | HCCEC | HCCEC from donor with ED | ED |
| GSM272859 | corpus cavernosum | HCCEC | HCCEC from donor without ED | Non_ED |
| GSM272860 | corpus cavernosum | HCCEC | HCCEC from donor with ED | ED |
| GSM272861 | corpus cavernosum | HCCEC | HCCEC from donor with ED | ED |
| GSM272862 | corpus cavernosum | HCCEC | HCCEC from donor with ED | ED |
| GSM272863 | Umbilical vein | HUVEC | HUVEC from donor without ED | Non_ED |
| GSM272864 | Umbilical vein | HUVEC | HUVEC from donor without ED | Non_ED |
| GSM272865 | Umbilical vein | HUVEC | HUVEC from donor without ED | Non_ED |
| GSM272866 | Coronary artery | HCAEC | HCAEC from donor without ED | Non_ED |
| GSM272867 | Coronary artery | HCAEC | HCAEC from donor without ED | Non_ED |
| GSM272868 | Coronary artery | HCAEC | HCAEC from donor without ED | Non_ED |
| GSM272870 | Coronary artery | HCAEC | HCAEC from donor without ED | Non_ED |

**Notes.**

ED, erectile dysfunction; non_ED, without erectile dysfunction.

Affymetrix Human Genome U133A 2.0 Array) (Table 1). This dataset contained four samples of human corpus cavernosum ECs from donors with ED, one sample of human corpus cavernosum ECs from donors without ED, and seven samples of human arteriovenous ECs from donors without ED (three human umbilical vein ECs and four human coronary artery ECs).

RNA quality was measured using the "RNA degradation map" provided by the Affy package in R language (4.0.0) (*Gautier et al., 2004*; *Wilson & Miller, 2005*). The dataset was standardized usung the limma package in R.

## Identification of differentially expressed genes (DEGs)

The dataset was classified into two groups: the ED group (four samples of human corpus cavernosum ECs from donors with ED) and the non-ED group (including one sample of human corpus cavernosum ECs from donors without ED, and seven samples of human arteriovenous ECs from donors without ED). The limma package in R was used with the Wilcoxon test to screen out the significant DEGs. According to the annotation information of GPL571 platform, the probes were transformed to the corresponding gene symbol. Probe sets without corresponding gene symbols were excluded and genes with more than one probe set were averaged. The cut-off values were set according to the parameters, an absolute logFC >0.5, and false discovery rate (FDR) < 0.05.

## Identification of autophagy-related DEGs (ARGs)

We obtained the autophagy-related genes (AREs) from GeneCard (https://www.genecards.org/). We entered the keywords "autophagy", used the relevance score >5.0 as the cutoff value, and downloaded them in the txt file format. Then, the DEGs and autophagy-related genes were entered into a Venn diagram (http://bioinformatics.psb.ugent.be/webtools/Venn/) to obtain the ARGs (*Wang, Thilmony & Gu, 2014*).

## Enrichment analysis and gene-concept network

We explored the correlation between all ARGs, up-regulated ARGs, down-regulated ARGs, and AREs using R. We focused on the functional enrichment analysis of Gene Ontology (GO), including the biological process, cellular component, and molecular function. We also analyzed the Kyoto Encyclopedia of Genes and Genomes (KEGG) of 20 ARGs.

We introduced the "cnetplot" to depict the linkages of genes and biological concepts of GO to form the Gene-Concept network (*Yu et al., 2012*).

## Interaction network of hub genes

The ARGs interaction network was created in STRING (http://string-db.org/cgi/input.pl) using the PPI network with the cutoff criteria as a combined score >0.4. Cytoscape (3.7.2) software was used to analyze and visualize the PPI network for hub genes with a cutoff value of >4.

## Construction of target gene-miRNA network and target gene-transcription factor (TF) network

The expression of target genes in the post-transcriptional stage was regulated by miRNA or TF under specific disease conditions. *Baldwin Jr, 2001*; *Liu et al., 2020*; *Soifer, Rossi & Saetrom, 2007*) NetworkAnalyst (https://www.networkanalyst.ca/) was used to analyze gene-TF interaction (ENCODE database) and gene-miRNA interaction (miRTarBase v8.0 database) (*Xia, Gill & Hancock, 2015*) and was visualized using Cytoscape.

## Animal and experiment designs

Eight-week old male Sprague-Dawley (SD) rats, weighing approximatley 200 g, with normal erectile function were provided by the Experimental Animal Center of Southern Medical University. The Institutional Research Ethics Committee of Nanfang Hospital of Southern Medical University provided full approval for this research (NFYY-2020-64). The 3R principle was used in our animal experiments: reduction of animal numbers, replacement of animals using other methods, and refinement of animal welfare. The experimental rats were randomly divided into the DMED group and the control group (NC). The rats with DMED were modeled and identified as previously mentioned (*Zhang et al., 2019b*). Briefly, healthy adult male SD rats were intraperitoneally injected with a 1% streptozotocin solution (65 mg kg$^{-1}$). Diabetes was determined by measuring random tail vein blood glucose levels 72 h after injection. Rats with random blood glucose concentrations >16.7 mmol L$^{-1}$ were diagnosed as diabetic. Six DMED rats were included in this study; six control rats were not treated. We used a blood glucose meter (Anwen, Changsha, China) to measure random blood glucose in the tail vein blood every two weeks. Weight was also evaluated every two weeks.

All rats were reared in the Specific Pathogen Free (SPF) temperature-controlled animal house of Nanfang hospital's animal center with a 12-hour light-dark cycle. A specialized breeder changed the padding once a day and fed the rats with warm water that had been boiled and ordinary food (produced by Yancheng Biotechnology Co., Ltd., Guangzhou, China).
## Erectile function evaluation

The mean ICP and ICP/MAP ratio were applied to evaluate erectile function, as mentioned above (*Zhang et al., 2017*). All rats were anesthetized with an intraperitoneal injection of sodium pentobarbital (30 mg/ kg). The cavernous nerves were completely exposed and the corpus cavernosum was carefully isolated. A 25-G needle containing a 100 U/ml heparin solution was slowly inserted into proximal corpus cavernosum to ensure that the rat was securely connected with the sensor and amplifier (MP150 Biopac System; Biopac Systems Inc., CA, USA). Both were connected using AcqKnowledge® (V4.4) software. We recorded the intracavernous pressure (ICP) when the cavernous nerve was stimulated with bipolar stainless steel electrodes. The mean atrial blood pressure (MAP) was measured by intubation in the biological signal system above after exposing the left carotid artery. A successful DMED rat model should meet an ICP of less than 60 mmHg and a ICP/MAP ratio less than 0.5. Data were recorded and the penis shaft was collected for future study. All rats were euthanized using a 10-fold anesthetic dose (300 mg/kg sodium pentobarbital).

## Histology

Freshly dissected tissue was fixated. H&E and Masson's trichrome staining was performed to determine tissue structure changes based on the manufacturer's instructions. We also determined the ratio between smooth muscle and collagen in the corpus cavernosum. Images were captured using Olympus BX63 microscopy (Olympus, Tokyo, Japan).

## Immunofluorescence analysis and laser confocal microscope

Tissue slides were prepared in a smiliar manner to those used in H&E staining. After antigen repair, goat serum was inoculated at room temperature for 30 min, the CD31 antibody (ab182981; Abcam) was incubated overnight, and a secondary antibody solution of Goat Anti-Rabbit IgG (HRP) (ab205718; Abcam) was added. Antigen repair was performed after the first dyeing and cleaning. Donkey serum was inoculated at room temperature for 30 min, the LC3 antibody (1:200; Proteintech, USA) was incubated overnight, and Donkey Anti-Rabbit 594 secondary antibody (Abcam, ab150075) was added through a drip. We used $4'$, 6-Diamidino-2-phenylindole (DAPI, #C0060; Solarbio) to stain the cell nuclei and images were captured using laser confocal microscopy (Nikon, Tokyo, Japan).

## Real-time quantitative PCR (qRT-PCR)

Cavernous tissue RNA was extracted and used for qRT-PCR analysis. The primer sequences we used were: $\beta$-Actin, sense: $5'$-GATCA-AGATCATTGCTCCTCCTG-$3'$, anti-sense: $5'$-AGGGTGTAAAACGCAG-CTCA-$3'$; NOTCH1, sense: $5'$-CAGTACAACCCGCTAAGGC, anti-sense: $5'$-GGACAAGG-TATTGGTGGAGA-$3'$; NOS3, sense: $5'$-CCGATTACACGAC-ATTGAGA-$3$ ', anti-sense: $5'$-TGGTCCAGTTGGG-AGCAT-$3'$ (anti-sense); KEAP1, sense: $5'$-GGTCGC-CCTGTGCCTCTAT-$3'$, anti-sense: $5'$-CACGCTGCTGTGGTGGAT-$3'$; CDKN2A, sense: $5'$-GAGGGCTTCCTAGACACTCTG- $3'$, anti-sense: $5'$-CGCAAATACCGCA-CGAC- $3'$; PPARG, sense: $5'$-CCTCCCTGATGAATAAAGA-$3'$, anti-sense: $5'$-AAC-TCAAACTTAGGCTCCA-$3'$; IRS1, sense: $5'$-GAGTGGTGGAGTT-GAGTTGG-$3'$, anti-sense: $5'$-GTCCGCATGTCAGCATA-$3'$.

## Western blot (WB)

WB was carried out for protein expression analysis as previously described (*He et al., 2012*). We detected LC3 and Beclin-1 using a rabbit anti-LC3 antibody (14600-1-AP) (1:1,000), anti-Beclin-1 antibody (11306-1-AP; Proteintech) (1:1,000), anti-NOTCH1 antibody (10062-2-AP; Proteintech) (1:1000), and anti-eNOS (Abcam, ab199956) (1:1000). Secondary antibodies were purchased from LI-COR Biosciences (catalog #D00825-14) and were diluted at 1:15,000. $\beta$-actin (AC026; ABclonal) was used as a loading control.

## Statistical analysis

The gray value and fluorescence intensity of image were further analyzed using Image J software. The outcomes were shown as the mean $\pm$ standard error. One-way ANOVA was applied to perform statistical comparisons among the groups. R software (version 4.0.0) and GraphPad Prism 8.3.0 were used to perform all statistical analyses. A *P*-value of <0.05 was considered statistically significant.

# RESULTS

## Identification of differentially expressed autophagy-related genes (ARGs)

We retrieved a total of four ED samples and eight non-ED samples with gene expression profiles from the GEO dataset. The RNA degradation plot was introduced to measure the quality of the RNA in the raw-data sample. The fluorescence intensity at the 5′end of the chip was much lower than that at the 3′end, and the slopes of the curves ranked in a similar order (Fig. 1A). We normalized the raw data for further analysis (Fig. 1B). We identified 1,236 significant DEGs between the ED and non-ED samples. Among these DEGs, 423 genes were up-regulated in ED tissue compared with non-ED tissue; the other 815 genes were down-regulated *via* Volcano Plot and heatmap (Fig. 1C). We matched DEGs with 380 AREs from the GeneCard (File S1) using a Venn diagram (Fig. 1D), resulting in 20 ARGs. Thirteen ARGs were down-regulated in ED tissues; the other seven genes were up-regulated compared with non-ED tissues (Table 2, Fig. 1E).

## Functional enrichment analysis and interactions of ARGs

All twenty ARGs were used for functional enrichment analysis to explore the biological functions and pathways of ARGs in autophagy. GO results showed that the biological process (BP) of ARGs were mainly enriched in autophagy, a process utilizing autophagic mechanism, the regulation of autophagy, positive regulation of proteolysis, and the positive regulation of the cellular catabolic process. The cellular component of ARGs was enriched in the nuclear outer membrane. ARGs' molecular functions were enriched with oxidoreductase activity, acting on NADH, heme protein as acceptor, enzyme inhibitor activity, Hsp70 protein binding, ligand-activated transcription factor activity, and nuclear receptor activity (Figs. 2A, 2C). The results of KEGG enrichment of ARGs were significantly enriched in the mTOR signaling pathway, PI3K-Akt signaling pathway, longevity regulating pathway, AMPK signaling pathway, and platelet activation, excluding autophagy-related pathways (Fig. 2B).

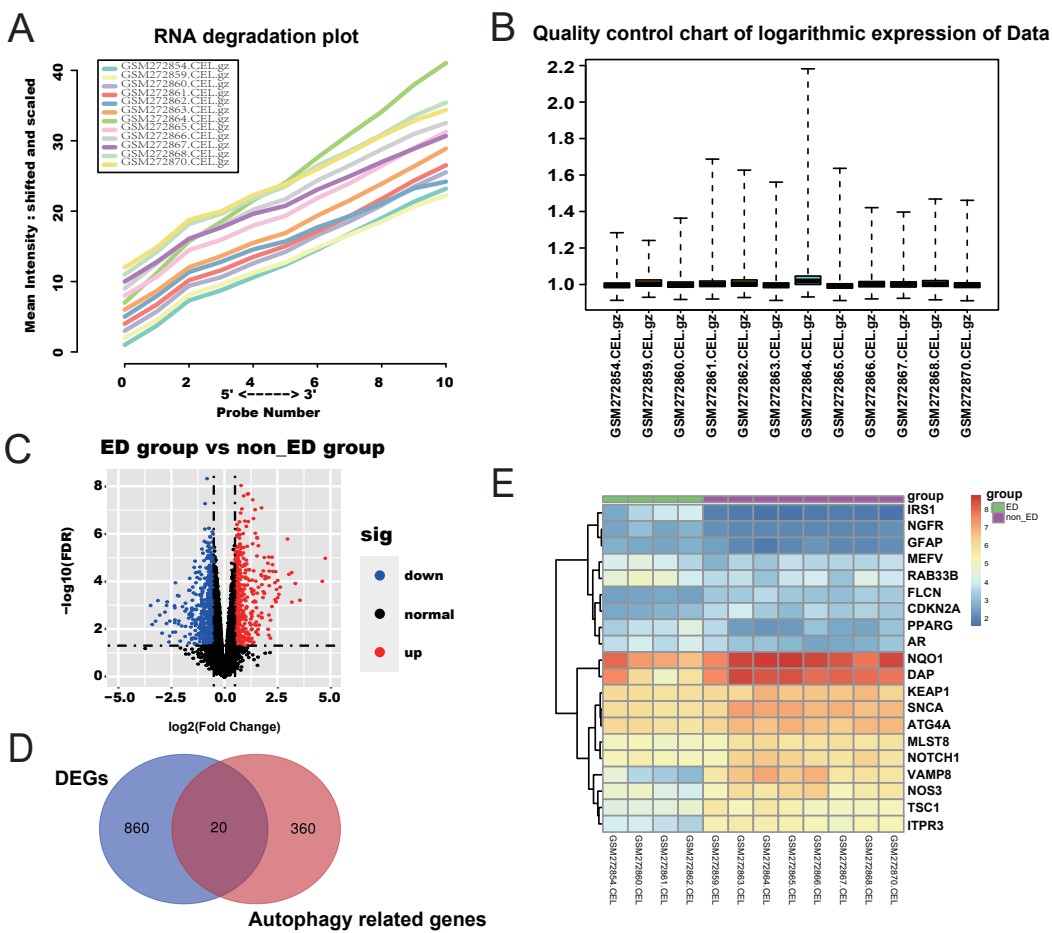

**Figure 1** **Checking of RNA Quality, normalizing of GEO dataset, and screening of differentially expressed autophagy-related genes (ARGs).** (A) RNA degradation plot. (B) Boxplot of the relative logarithmic expression (RLE) reflected the consistency of parallel experiments. (C) Volcano plots showed the differentially expressed genes (DEGs) from GSE10804. Data points in red represent down-regulated, and green represent up-regulated genes. (D) Venn diagram was performed to screen the ARGs from the DEGs and autophagy-related genes. (E) Heatmap showed the 20 ARGs expression in ED and non-ED groups.

The enriched GO pathways of molecular functions of seven up-regulated ARGs were primarily nuclear receptor activity, ligand-activated transcription factor activity, and steroid hormone receptor activity (Fig. 2D). GO enrichment of 13 down-regulated ARGs showed that molecular functions were enriched in oxidoreductase activity, acting on NAD(P)H, heme protein as acceptor, enzyme inhibitor activity, and Hsp70 protein binding (Fig. 2E). The GO enrichment of 380 AREs showed that molecular functions were enriched in ubiquitin-protein ligase binding, ubiquitin-like protein ligase binding, protein serine/threonine kinase activity, phosphatase binding, protein phosphatase binding, and heat shock protein binding (Fig. 2F). Interestingly, heat shock protein (HSP) binding appeared in three groups.

**Table 2  Significantly different expressed levels of ARGs in ED and non-ED tissues.**

| Gene | logFC | AveExpr | t | P.Value | adj. P. Val | B |
|------|-------|---------|---|---------|-------------|---|
| MEFV | 0.583129 | 3.692823 | 3.142959 | 0.008025 | 0.046334 | −2.76814 |
| VAMP8 | −2.33546 | 5.195704 | −4.11645 | 0.001289 | 0.016228 | −0.97383 |
| RAB33B | 0.848874 | 4.013352 | 3.834705 | 0.002174 | 0.021723 | −1.48961 |
| IRS1 | 1.483492 | 2.269639 | 5.593557 | 9.72E−05 | 0.003723 | 1.587609 |
| PPARG | 1.056873 | 3.134478 | 4.170111 | 0.001168 | 0.015245 | −0.8763 |
| NGFR | 0.670848 | 2.323239 | 7.183001 | 8.39E−06 | 0.001237 | 3.993815 |
| SNCA | −0.85481 | 6.485949 | −5.42034 | 0.00013 | 0.004494 | 1.30249 |
| KEAP1 | −0.51936 | 6.318866 | −3.9344 | 0.001805 | 0.019673 | −1.30647 |
| ATG4A | −0.56943 | 6.423723 | −5.12685 | 0.000213 | 0.005868 | 0.809451 |
| FLCN | −0.75199 | 3.080793 | −7.52722 | 5.16E−06 | 0.001042 | 4.465874 |
| TSC1 | −0.67593 | 4.957076 | −4.42259 | 0.000737 | 0.011674 | −0.42107 |
| NQO1 | −1.12484 | 7.957864 | −4.21921 | 0.001067 | 0.014561 | −0.78729 |
| MLST8 | −0.7294 | 5.552902 | −5.46794 | 0.00012 | 0.004307 | 1.381291 |
| NOS3 | −1.13072 | 5.27702 | −3.47571 | 0.004269 | 0.032331 | −2.15277 |
| GFAP | 0.598847 | 2.32642 | 5.327376 | 0.000152 | 0.004764 | 1.147666 |
| DAP | −0.53841 | 7.869929 | −3.34197 | 0.0055 | 0.037284 | −2.40038 |
| CDKN2A | −0.54667 | 3.220725 | −3.4103 | 0.004831 | 0.034619 | −2.27388 |
| AR | 0.879528 | 3.323439 | 4.801902 | 0.000375 | 0.008083 | 0.249554 |
| NOTCH1 | −0.78118 | 5.779909 | −3.66608 | 0.002981 | 0.026172 | −1.80058 |
| ITPR3 | −1.35676 | 4.747004 | −6.09948 | 4.29E−05 | 0.002502 | 2.394674 |

**Notes.**

LogFC, log fold change; adj.P.Val, adjust P value.

## Construction of the PPI network, gene-microRNA network, and gene-TF network

We explored the twenty ARGs interactions using the STRING website (Fig. 3A). The gene network demonstrated that IRS1, CDKN2A, NOTCH1, PPARG, KEAP1, and NOS3 were the hub genes using Cytospace software (Fig. 3B). The top three ARGs for miRNAs were AR, NGFR, and FLCN. Hsa-mir-24-3p and Hsa-mir-335-5p may control the largest number of ARGs (Fig. 3C). NOS3, VAMP8, and MLST8 were the top three target ARGs for TFs, modulated by 78, 77, and 70 TFs (Fig. 3D).

## Blood glucose, weight, ICP/MAP, and morphological changes of penis corpus cavernosum of DMED model

The blood glucose of DMED rats was significantly higher than that of the NC rats, while the weight of DMED rats was significantly lower than that of the NC rats at the 8th week after modeling (Fig. 4A). We successfully established the DMED rat model with ICP less than 60 mmHg and ICP/MAP ratio less than 0.5 (Fig. 4B). Results from H&E and Masson's trichrome staining revealed that the corpus cavernosum was disordered and the ratio of fibroblast to collagen decreased when compared with the NC group. The endothelium and smooth muscle layer in the DMED group were thinner than the NC group (Figs. 4C–4D).

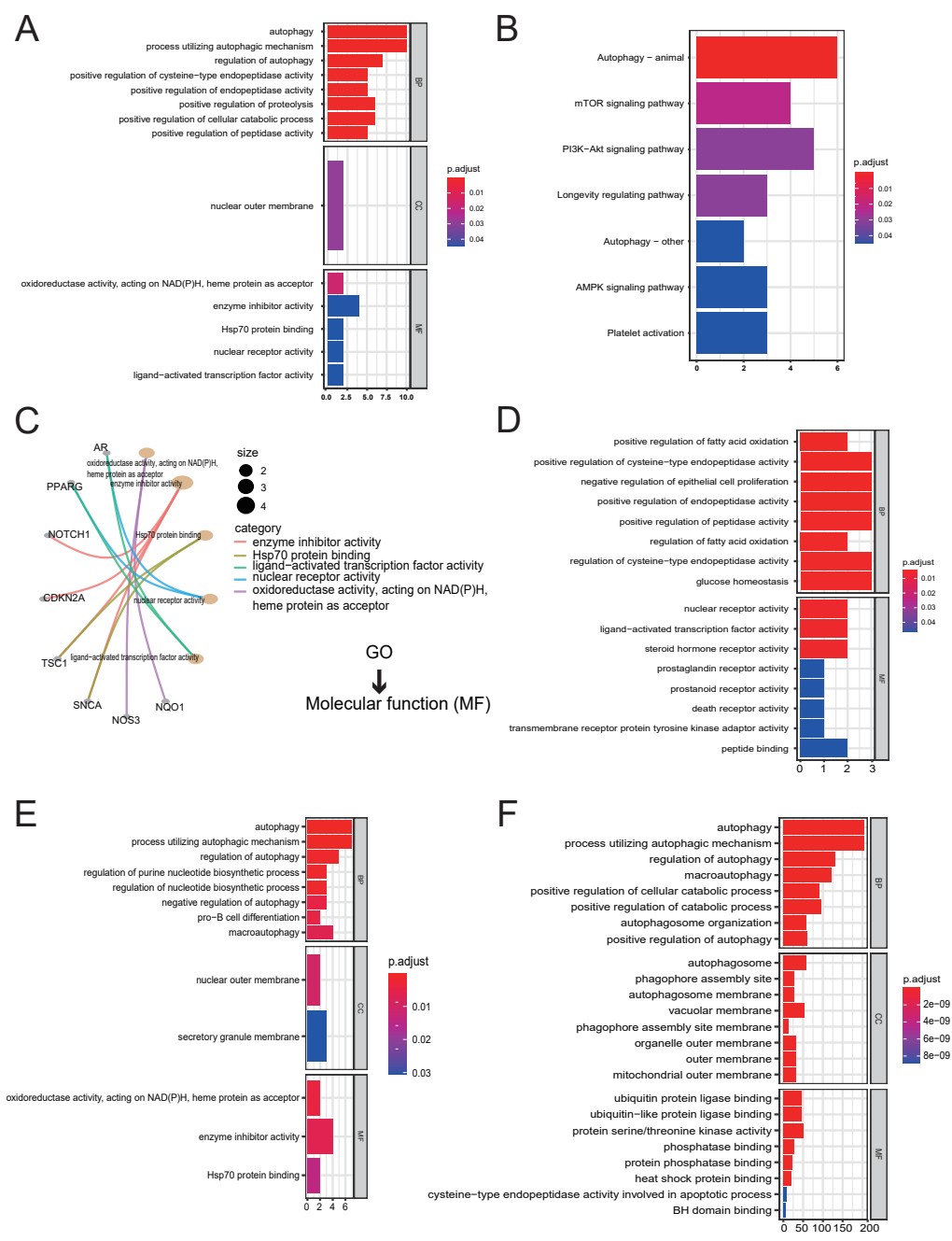

**Figure 2** **Gene Ontology (GO) enrichment analysis and KEGG pathway analysis.** (A) GO enrichment significance items of 20 ARGs were shown with bar plot in three functional groups: molecular function (MF), biological processes (BP), and cell composition (CC). The x-axis label represents the gene ratio, and the y-axis label represents GO terms. (B) KEGG pathway analysis was performed with bar plot. (C) The centplot for the MF data in GO analysis was shown. (D) GO enrichment significance items of seven up-regulated ARGs were shown with bar plot in two functional groups: MF and BP. The x-axis label represents the gene ratio, and the y-axis label represents GO terms. (E) GO enrichment significance items of 13 down-regulated ARGs were shown with bar plot in three functional groups: MF, BP, and CC. The x-axis label represents the gene ratio, and the y-axis label represents GO terms. (F) GO enrichment significance items of 380 autophagy genes were shown with bar plot in three functional groups: MF, BP, and CC. The x-axis label represents the gene ratio, and the y-axis label represents GO terms.

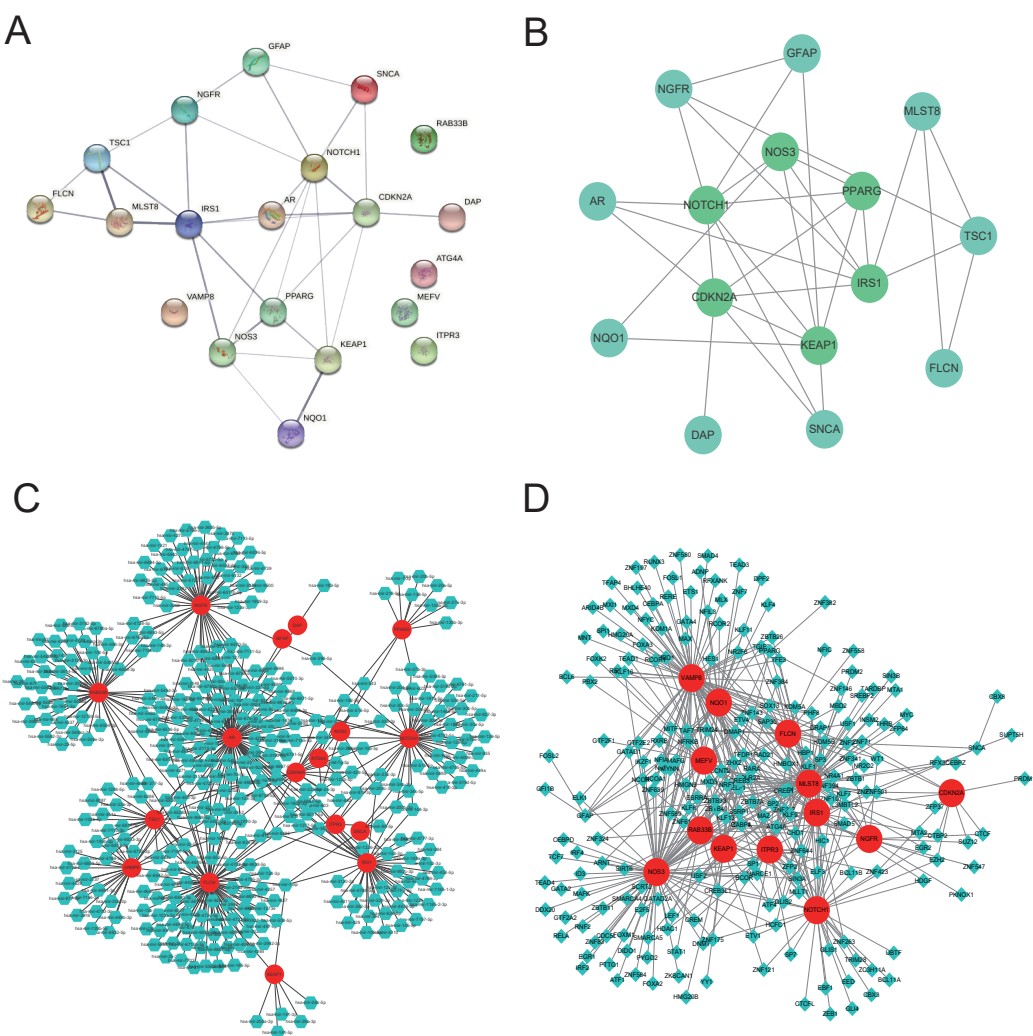

**Figure 3  Gene interactions and enrichment of ARGs.** (A) The gene network downloaded from the STRING database indicated the interactions among the ARGs. (B) PPI network of ARGs by Cytoscape. Node color reflects the degree of connectivity (dark-green color represents a higher degree, and light-green color represents a lower degree)—networks of (C) target gene-miRNA and (D) target gene-TF. The red circle nodes are the genes, green hexagon nodes are the miRNAs, and green diamond nodes are the TFs.

## Enhanced autophagy of cavernous tissue and Endothelium in DMED model

The protein level of LC3-II and LC3-I, and the ratio of LC3-II/LC3-I were upregulated, and the grayscale value of Beclin-1 was downregulated in DMED group compared with the NC group (Fig. 5A). The fluorescence intensity of LC3 in the DMED group was significantly higher than that in the NC group. The increase of LC3 represented an increase in autophagosomes. The discontinuity of CD31 labeled ECs, the decrease of fluorescence intensity, and the disappearance of fluorescence on smooth muscle surface suggested that the ECs in DMED tissue were injured (Fig. 5B).

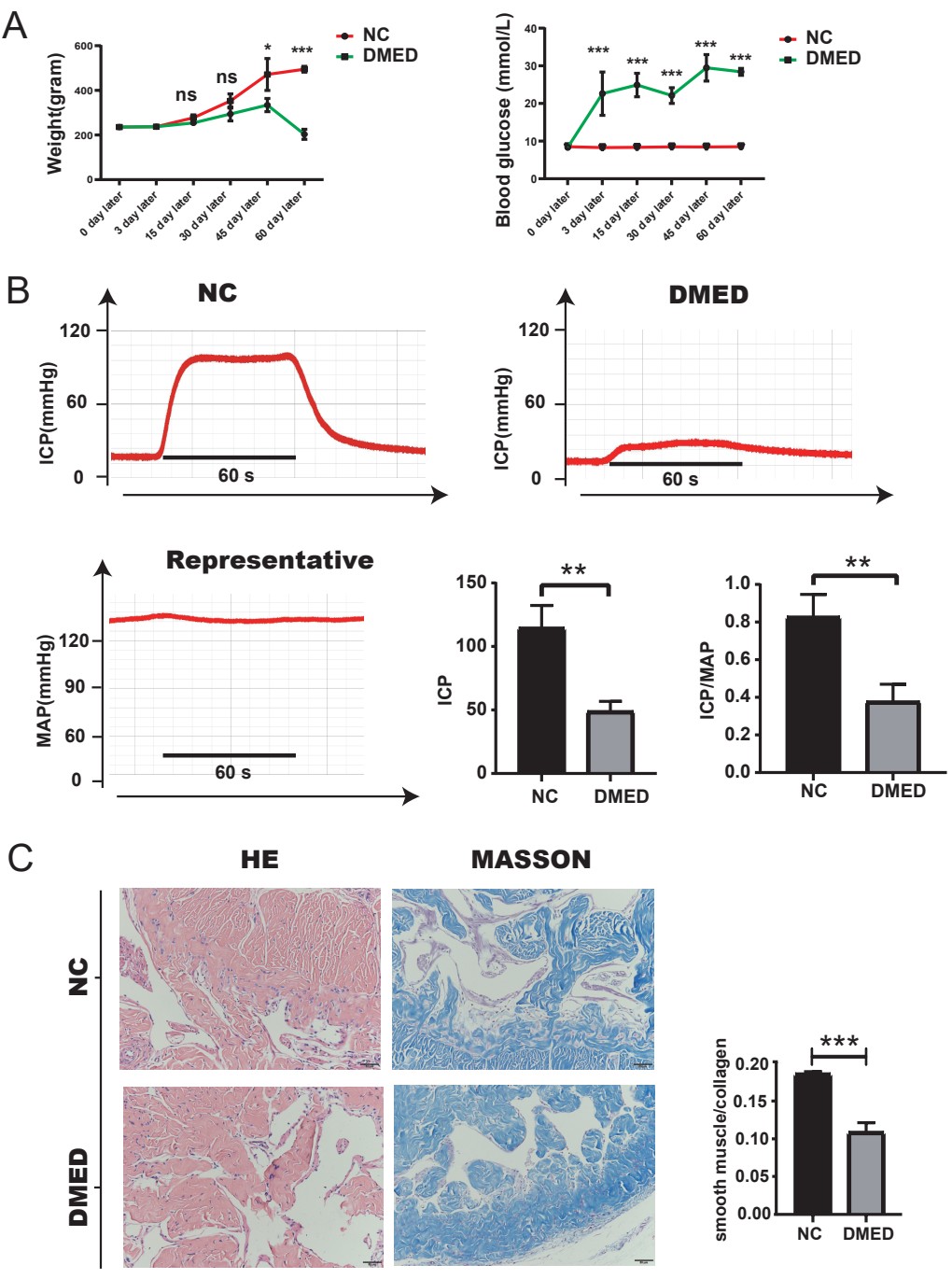

**Figure 4   Establishment of DMED rat model.** (A) Bodyweight after 8 weeks' feeding. (B) blood glucose level in each group. Each bar represents mean ± SEM. (B) Results of erectile function expressed as ICP and the ratio of ICP/MAP. Erectile function decreased significantly in the DMED group. Expressed as ICP/MAP ratio. (C) and (D) H&E and Masson's trichrome staining showed significant differences in morphology and Smooth Muscle-to-collagen ratio among the different groups. $n = 3$ rats per group. * $p < 0.05$, ** $p < 0.01$, *** $p < 0.001$.

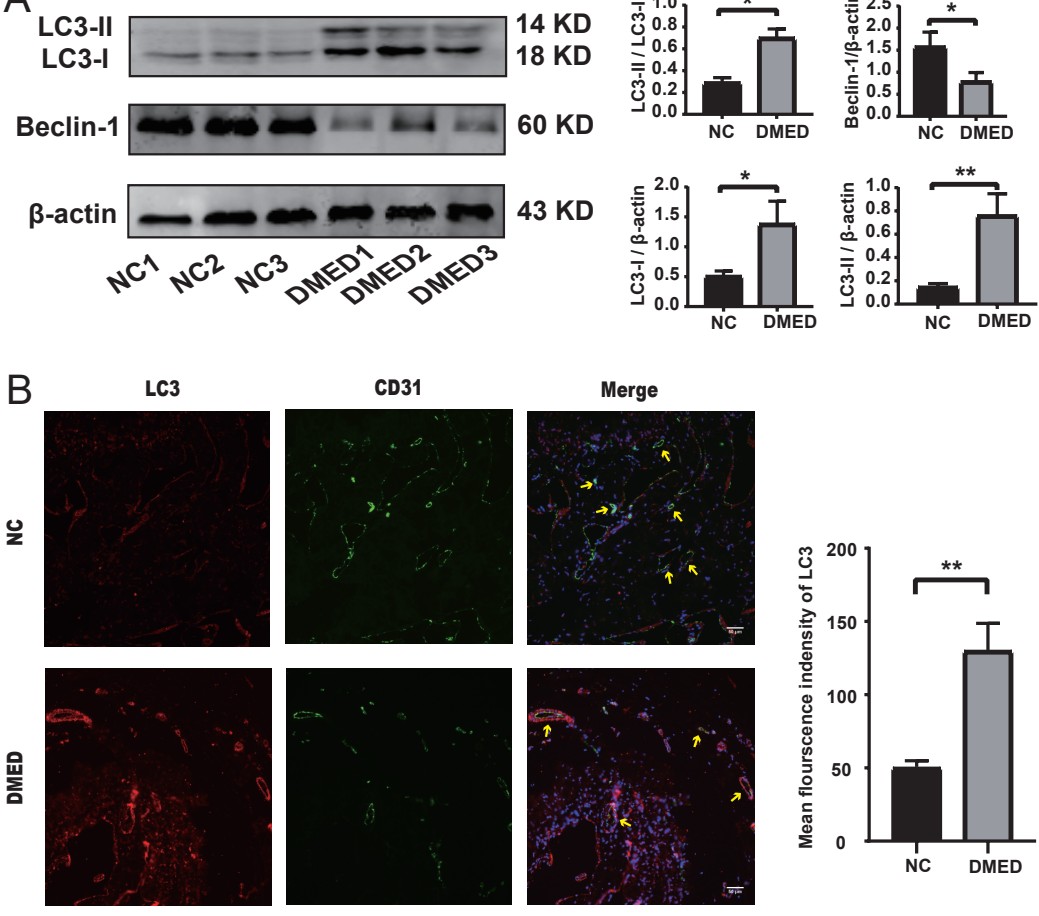

**Figure 5** **Autophagy increased, and endothelium was injured in the DMED rat model.** (A) Western blot showed that significantly increased LC3II, LC3I, LC3II/LC3I ratio, and reduced Beclin-1 expression were detected in DMED groups. (B) Immunofluorescence staining for CD31 and LC3 in vascular endothelial cells of corpus cavernosum in each group (yellow arrow points to the sinus). Animals tested: $n = 6$. $^*p < 0.05$.

## The expression level of hub ARGs

The results of qRT-PCR showed that the expressions of Notch1, PPARG, NOS3, CDKN2A, and IRS1 were down-regulated, however, there was no statistical significance in the expression of KEAP1 compared to NC group (Figs. 6A–6F). The gene expression of NOTCH1, CDKN2A, and NOS3 was consistent with the data in Table 2. The protein level of NOTCH1, CDKN2A, and NOS3 were downregulated in DMED group compared to NC (Fig. 6G).

## DISCUSSION

Endothelial dysfunction is one of the leading causes of ED. ECs play a critical role in regulating inflammation, platelet aggregation, vascular smooth muscle hyperplasia, and thrombosis. Thus, targeting endothelial dysfunction may be a more effective treatment for

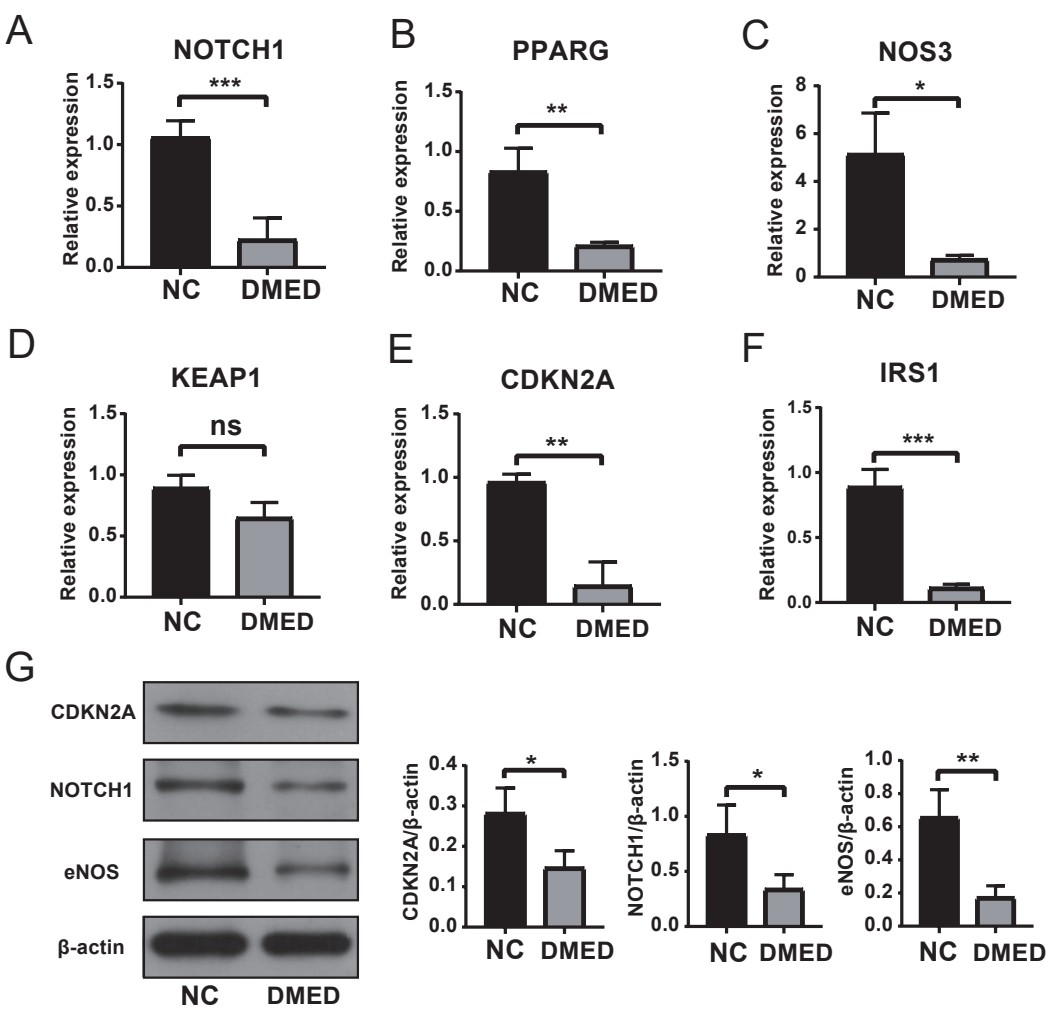

**Figure 6** **Identification of Hub ARGs with DMED model by qRT-PCR and WB.** (A) expression of NOTCH1, (B) expression of PPARG, (C) expression of NOS3, (D) expression of KEAP1, (E) expression of CDKN2A, (F) expression of IRS1, (G) protein level of CDKN2A, NOTCH1, and eNOS. ns: No significancy, $^*p < 0.05$, $^{**}p < 0.01$, $^{***}p < 0.001$.

ED. Studies have suggested that endothelial dysfunction may be alleviated by inhibiting autophagy (*Altabas & Altabas, 2015*; *Verma & Anderson, 2002*). Autophagy is a multi-step collaborative process regulated by autophagy-related genes (*Sarkar et al., 2014*). However, the pathway of autophagy-related genes and ED remains unclear. In this study, we utilized bioinformatics to mine twenty autophagy-related genes and their related molecular functions in an ED database. Moreover, six hub genes were further screened out using the PPI co-expression network. We further verified these genes by constructing rat model of diabetic ED relying on the homology of human and mammalian genes.

The LC3-II/LC3-I ratio and LC3-II/ $\beta$-actin ratio in the DMED group was relatively higher than the control group. LC3-II is bound to the cell membrane in mature autophagosomes, is released upon fusion with the lysosome, and is considered to be a

marker for autophagy detection (*Ni et al., 2011*). This suggests that autophagy is enhanced in DMED tissue. In addition, the enhanced autophagy of endothelial cells was confirmed by CD31 and LC3 co-staining. Excessive ROS accumulation promotes compensatory autophagy during the pathogenesis of diabetic ED, which protects the body from oxidative stress by clearing damaged intracellular substances (*Scherz-Shouval et al., 2007*). The Beclin1/ $\beta$-actin ratio in the DMED group was lower than in the control. This protein plays a central role in the formation and maturation of autophagosomes. The conditions that disrupt or promote the Beclin1-Bcl2 complex play a crucial part in determining whether cells undergo autophagy or apoptosis. A recent study showed that when Beclin1 was induced it facilitated autophagy in various cell types, including ECs (*Leonard et al., 2019*). We found that the gene expression of NOTCH1, CDKN2A, and NOS3 was consistent with the high throughput sequencing results from GEO. These may become underlying target molecules in the regulation of EC autophagy in ED.

Notch is a classical signaling pathway, including four Notch receptors (Notch 1, 2, 3 and 4) as well as five notch ligands (delta-like receptors 1, 3, 4 and jagged 1 and 2) in mammals (*Gordon, Arnett & Blacklow, 2008*). The pathway is activated when a Notch receptor interacts with a ligand. Studies suggested that the inhibition of autophagy leads to the activation of Notch pathway. Autophagy suppresses the Notch1 pathway by up-regulating the degradation of Notch1 (*Li et al., 2016*; *Wu et al., 2016*; *Zeng et al., 2016*). The reduction of NOTCH1 in ECs is a predisposing factor for vascular inflammation and atherosclerosis (*Briot et al., 2015*). Endothelial dysfunction may be associated with a decrease of Notch1. Moreover, the Notch pathway was associated with senescence which the mainstream factor of ED. A recent study showed that the Notch-1 inhibitor, Tangeretin, alleviated ED by increasing the maximum ICP/MAP and up-regulating the expression of eNOS in hypertensive rats (*Chiangsaen et al., 2020*). Thus, NOTCH1 may be a candidate biomarker for ED.

CDKN2A, also called p16$^{INK4a}$ (p16), has been shown to increase with age in several rodent and human tissues (*Baker et al., 2011*). As a member of the INK4 family of cyclin-dependent kinase inhibitors, p16 plays a critical role in cell-cycle regulation. p16 expression inhibits cellular proliferation by downregulating cyclin-dependent kinases 4 and 6 (CDK4/6) (*Serrano, 1997*). As a tumor suppressor, p16$^{INK4a}$ (p16) is a well-studied maturation marker, inducing cellular senescence (permanent cell-cycle arrest) in response to stress. p16 accumulates in tissues with age, causing cellular senescence. The elimination of p16 expression in senescent cells is associated with a prolonged life span and a reduction of tumorigenesis (*Baker et al., 2011*). Studies have suggested that autophagy inhibitors bafilomycin A1 and chloroquine induced p16 accumulation in stagnant vesicles containing the autophagy marker LC3. The accumulation of p16 in these vesicles was consistent with the increase of LC3-II. The knockout of autophagosome chaperone p62 attenuated p16 aggregates in lysosomes, indicating that p16 targets these vesicles through p62. As the regulator of autophagy, p16 performs a key role in the etiology of cancer and dementia (*Coryell et al., 2020*). Moreover, advanced age is a critical risk factor for most chronic diseases and functional defects in humans. A significant increase has been shown in ED incidence in men over the age of 40 (*Shamloul & Ghanem, 2013*). Senescence is closely

associated with ED. We found that the expression of CDKN2A was decreased. This result suggests that not all cells with high expression of p16$^{Ink4a}$ are senescent cells and not every senescent cell has high expression of p16$^{Ink4}$ (*Hall et al., 2017*). There should be further studies conducted on CDKN2A as an autophagy-related molecule.

Endothelial nitric oxide synthase (eNOS) is also known as NOS3, one of three subtypes of nitric oxide synthase (NOS). It is involved in the synthesis of nitric oxide (NO) with L-arginine and molecular oxygen as substrates. It is also involved in the regulation of physiological and pathological functions. The pathophysiology of endothelial dysfunction involves multiple mechanisms, such as the dysregulation of NO by vascular/endothelial eNOS in ED.

eNOS performs an essential role in erectile response. eNOS activity and endothelial NO bioavailability in the penis are jointly regulated by multiple post-translational molecular mechanisms, including eNOS phosphorylation, eNOS interaction with upstream proteins and contractile pathways, and the generation of reactive oxygen species (ROS). The activation of the PI3K/Akt/eNOS pathway was considered to be one of classic pathways to regulate endothelial dysfunction and participate in the cellular and molecular biology of ECs (*Chen et al., 2021*; *Chen et al., 2019*; *Duan et al., 2019*; *Zhang & Zhang, 2020*). The regulation of the AMPK pathway and the mTOR pathway corresponded to endothelial autophagy (*Xiong et al., 2014*). Our results confirmed that autophagosome increased in the cavernous ECs of DMED rats, suggesting an increase in autophagy. The participation of eNOS in the autophagy pathway in endothelial dysfunction requires further experimental study.

Previous literature reporting on mammalian cells has indicated that protein degradation pathways are commonly classified in the ubiquitin-proteasome system. There are three main types of autophagy: macro-autophagy, micro-autophagy, and chaperone-mediated autophagy. Hsp70 acts a central hub in protein degradation through the ubiquitin-proteasome system and different autophagy pathways (*Fernández-Fernández et al., 2017*). Our GO analysis suggested downregulated ARGs and 380 AREs were both associated with HSP protein binding. The Hsp70 family performs central roles in every aspect of proteostasis, from protein folding to disaggregation and degradation (*Mayer & Bukau, 2005*). Targeting HSP70 binding in ED may be a new direction for future research. Furthermore, the upregulated ARGs (AR and PPARG) were mainly enriched in nuclear receptor activity and downregulated ARGs (TSC1 and SNCA) were mainly enriched in oxidoreductase activity. However, their relationship with ED requires further study.

The top three targeted ARGs in the miRNA-gene network were AR, NGFR, and FLCN. It has been suggested that nerve growth factor (NGF) may induce nerve regeneration by activating the autophagy of Schwann cells. It may also promote the maintenance, survival, and function of vascular endothelial cells (*Tanaka et al., 2004*). Low doses of rapamycin, an autophagic inducer, accelerated autophagy by activating the NGFR promoter. Therefore, NGFR may be involved in the regulation of endothelial autophagy. Hsa-mir-24-3p and Hsa-mir-335-5p, which control more ARGs, perform a key role in angiogenesis (*Esquinas et al., 2017*; *Li et al., 2020*). The relationship between AR, FLCN, and endothelial dysfunction requires further study. NOS3, VAMP8, and MLST8 were ranked as the top three targeted

ARGs in the TF-gene network. A previous study showed that platelets stimulated BMDC homing and promoted angiogenesis. Thus they are a key mediator between hypoxic tissue and bone marrow, in which VAMP8 plays an important role (*Feng et al., 2011*). In endothelial cells, MLST8 is involved in modulating the activation of Akt and reducing the phosphorylation of Akt targets such as the eNOS and FOXO subfamily (*Partovian et al., 2008*). The role of NOS3 in endothelial dysfunction has been discussed.

Recently there has been great interest in the relationship between autophagy and ED. Glucagon-like peptide-1 (GLP-1) receptor agonists, defocused low-energy shock wave therapy, and stem cell therapy have been suggested to alleviate ED by promoting endothelium autophay in DMED rats (*Yuan et al., 2020*; *Zhang et al., 2019a*; *Zhu et al., 2018*). Autophagy was shown in the ED model of BCNI and aging. Thus ED may be allieviated by improving autophagy (*Tang et al., 2018*; *Ye et al., 2021*). However, there is no research on the expression changes and molecular functions of ED-related autophagy genes. Our study may provide new directions for future research and novel molecular targets for the treatment of ED.

Our research was limited by the small number of microarray samples which may lead to statistical bias. Secondly, we screened genes based on microarray data sourced from human tissue in GEO. However, this was verified on SD rats which may cause species bias. We have also not specifically studied whether these genes can regulate autophagy.

## CONCLUSIONS

Our study mined endothelial-related ARGs to identify the possible pathogenesis of ED related to autophagy. Consistent with the fact that high glucose accelerates the senescence of endothelial cells by inhibiting autophagy, two of three identified genes, NOTCH1 and CDKN2A, are involved in senescence. NOTCH1, CDKN2A, and NO3 may be the underlying therapeutic targets to treat ED. Moreover, the study of upstream transcription factors and miRNAs of target genes may improve the mechanistic understanding of endothelial dysfunction in ED.

### Funding
This work was supported by the National Natural Science Foundation of China (82060809). The funders had no role in study design, data collection and analysis, decision to publish, or preparation of the manuscript.

### Grant Disclosures
The following grant information was disclosed by the authors:
National Natural Science Foundation of China: 82060809.

### Competing Interests
The authors declare there are no competing interests.
## Author Contributions

- Chao Luo conceived and designed the experiments, performed the experiments, authored or reviewed drafts of the paper, and approved the final draft.
- Xiongcai Zhou performed the experiments, authored or reviewed drafts of the paper, and approved the final draft.
- Li Wang and Qinyu Zeng analyzed the data, authored or reviewed drafts of the paper, and approved the final draft.
- Junhong Fan and Shuhua He analyzed the data, prepared figures and/or tables, and approved the final draft.
- Haibo Zhang and Anyang Wei conceived and designed the experiments, authored or reviewed drafts of the paper, and approved the final draft.

## Animal Ethics

The following information was supplied relating to ethical approvals (i.e., approving body and any reference numbers):

The Institutional Research Ethics Committee of Nanfang Hospital of Southern Medical University provided full approval for this research (NFYY-2020-64).

## Data Availability

Data are available at NCBI GEO: GSE10804. Raw measurements are available in the Supplemental Files.

## Supplemental Information

Supplemental information for this article can be found online at http://dx.doi.org/10.7717/peerj.11986#supplemental-information.

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
