# Peer review of "Screening and identification of NOTCH1, CDKN2A, and NOS3 as differentially expressed autophagy-related genes in erectile dysfunction"

_PeerJ, doi:10.7717/peerj.11986_

## Round 0.1 · original submission · Major Revisions

As you can see, all reviewers raised serious concerns that have to be addressed during revision.

Reviewer 1 ·

Basic reporting

The article must be written in English and must use clear, unambiguous, technically correct text.

Experimental design

no proper Experimental design

Validity of the findings

no clear conclusion is given

Additional comments

Author performed reanalyzed work
GEO accession no GSE10804 was previously analysed, work done and published.
1.Screening and identification of critical biomarkers in erectile dysfunction: evidence from bioinformatic analysis (PMID: 32161689 PMCID: PMC7050549 DOI: 10.7717/peerj.8653)
2. Transcriptional profiling of human cavernosal endothelial cells reveals distinctive cell adhesion phenotype and role for claudin 11 in vascular barrier function (PMID: 19622796 PMCID: PMC2765067 DOI: 10.1152/physiolgenomics.90354.2008)


Re analysed work is not acceptable Meanwhile, Author not provided Differential gene expression (DEGs) table with probe id, logFC, pValue, adj.P.Val, t value and Gene Name, which are more fundamental and basic in this work. No evedance in this work without this DEGstable.
Author not performed construction and analysis of target genes - TF regulatory network and target genes - miRNA regulatory network

Reviewer 2 ·

Basic reporting

The authors have written the manuscript with a good flow to address the hypothesis at hand

Experimental design

The experiment design is validated with enough number of biological replicates and statistical analysis

Validity of the findings

The results have been obtained from previously published data and have enough replicates to address statistically relevant information. The findings are then validated in animal model of the disease. They have been validated using qPCR and western blot approaches. However, there is still some information that needs to be added to strengthen the manuscript.

Additional comments

The authors have done a good job at putting the data together as well as at extracting the relevant data from the previous datasets. However, there are some key points that need to be addressed and have been added as comments in the reviewed version.
Please make sure to add better representative images for the IF staining.

Annotated reviews are not available for download in order to protect the identity of reviewers who chose to remain anonymous.

Reviewer 3 ·

Basic reporting

Overall the manuscript needs more work to improve the readability, e.g. more professional English descriptions and clearer expression:

Line 41-42: Analyzing GEO seems a bit vague, at least list the type and number of samples used.

Line 49: 20 ARGs identified ultimately is a bit unclear - this seems not a proper summary of the whole Results section.

Line 54-56; Line 85-87 : Grammer and sentence structure issues.

Line 105: Here it says 3 samples for the ED group, but later in Line 204 it says 4 samples.

Experimental design

1. I have some concerns on the GSE10804 dataset used in this project, which only contains 4 ED vs. 8 non-ED samples. Usually the DEGs identified from such a small sample size are prone to false positives.
2. Line 215m, after 20 ARG DEGs were identified, only 11 DEGs were used for enrichment analysis, is there a reason for only using a subset?

Validity of the findings

The autophagy related genes (ARGs) used in this manuscript lacks a clear definition. According to line 114, 381 ARGs were pulled from GeneCard by searching with the keyword "autophagy". However, the keyword "autophagy" gives us 6,346 results from GeneCard when I tried to replicate the procedure.

Additional comments

The therapeutic biomarkers has great values, and I believe this study will be of interest to scientists and clinicians involved in ED research in general. There are some areas the authors could modify to improve readability and clarity, as outlined above.

---

## Round 0.2 · accepted · Accept

Thank you for addressing the critiques of the reviewers. Your revised manuscript is acceptable now.

Reviewer 2 ·

Basic reporting

The reporting for the manuscript has improved through revisions

Experimental design

The experimental design has been updated and clarified by the authors to support their hypothesis

Validity of the findings

They have updated the manuscript based on the reviewers' suggestions